# Experimental Study of Body-Fin Interaction and Vortex Dynamics Generated by a Two Degree-Of-Freedom Fish Model

**DOI:** 10.3390/biomimetics4040067

**Published:** 2019-10-08

**Authors:** Seth A. Brooks, Melissa A. Green

**Affiliations:** Department of Mechanical and Aerospace Engineering, Syracuse University, Syracuse, NY 13244, USA; greenma@syr.edu

**Keywords:** biological fluid dynamics, bio-propulsion, fish, swimming, body-fin interaction, vortex dynamics, leading edge vortices, circulation production, nonsinusoidal motion

## Abstract

Oscillatory modes of swimming are used by a majority of aquatic swimmers to generate thrust. This work seeks to understand the phenomenological relationship between the body and caudal fin for fast and efficient thunniform swimming. Phase-averaged velocity data was collected and analyzed in order to understand the effects of body-fin kinematics on the wake behind a two degree-of-freedom fish model. The model is based on the yellowfin tuna (*Thunnus albacares*) which is known to be both fast and efficient. Velocity data was obtained along the side of the tail and caudal fin region as well as in the wake downstream of the caudal fin. Body-generated vortices were found to be small and have an insignificant effect on the caudal fin wake. The evolution of leading edge vortices formed on the caudal fin varied depending on the body-fin kinematics. The circulation produced at the trailing edge during each half-cycle was found to be relatively insensitive to the freestream velocity, but also varied with body-fin kinematics. Overall, the generation of vorticity in the wake was found to dependent on the trailing edge motion profile and velocity. Even relatively minor deviations from the commonly used model of sinusoidal motion is shown to change the strength and organization of coherent structures in the wake, which have been shown in the literature to be related to performance metrics such as thrust and efficiency.

## 1. Introduction

A majority of aquatic animals use oscillations of their body and fins to produce thrust for locomotion. The varying geometries and kinematics of these animals influence performance metrics such as efficiency, power, and maneuverability. The connections among these have been extensively investigated through various simplifications of the system. There are numerous forms of oscillatory motions that are used in aquatic swimming; however, thunniform swimmers are most commonly studied for their ability to swim efficiently and quickly [1,2,3]. This form of swimming is characterized by a relatively stiff body with the majority of lateral motion restricted to the posterior 10% of the body, which consists of the peduncle region and caudal fin [2]. The peduncle is the narrowest part of the body located in the posterior section and connects the main part of the body to the caudal fin, the most posterior fin. The fact that the lateral motion is concentrated in the posterior portion of the body has led many researchers to eliminate the complexity that arises from fin-fin and body-fin interaction by studying the caudal fin in isolation. In these cases, the caudal fin is typically modeled as a relatively simple pitching and/or heaving airfoil or thin plate, both rigid and flexible, in experiments and simulations. Several recent numerical works, however, have modeled the entire body of fish to understand thrust generation.

Early experimental work simplified the system to a symmetric rigid airfoil undergoing pure pitching, such as the work of Koochesfahani [4]. In the following years, researchers began using Strouhal number, St=fA/U, as the principal nondimensional parameter governing oscillatory propulsion where *f* is the pitching frequency; *A* is the width of the wake (maximum excursion of the trailing edge is commonly used); and *U* is the freestream velocity. Heave was later added to the motion by both Triantafyllou et al., and Read et al. [5,6]. Buchholz and Smits extended the two-dimensional understanding by investigating finite aspect ratio propulsors using a rectangular panel pitched about its leading edge [7]. Green et al., and King et al., took this a step closer to its bio-inspiration by modeling the isolated propulsor as a trapezoidal panel [8,9]. In the past decade, the field has divided into two groups that focus on either two-dimensional models for parametric studies or three-dimensional models with fewer cases. Within the two-dimensional group, there has been a recent push to understand fish propulsion through nondimensional scaling with a symmetric airfoil representing the caudal fin [10,11,12,13] or a flexible thin foil representing the undulating fish body [14,15,16]. Within the three-dimensional group, there are physical systems that mimic the motions of fish [17,18,19] as well as numerical works that investigate the fluid dynamics around full fish models [20,21,22,23,24].

Due to the complexity of recreating fish kinematics, much of the work in understanding the interaction between the median fins (dorsal, ventral, and caudal fins) has been done by experimental biologists [19,25,26,27,28]. Several studies on live fish suggest that the spacing between the dorsal/ventral fin and the caudal fin is significant when understanding the flow interaction between them [25,26,27,28]. Mignano et al., used a physical model to show that the distance between the median fins as well as the phase offset between the dorsal/ventral fin and caudal fin pitching motion can significantly affect thrust production [19]. Other studies use two-dimensional inline tandem foils to simulate the interaction between fins [29,30,31,32]. They all show that with proper phase offset and physical spacing between the two foils, the performance of the hind foil (caudal fin) can be substantially enhanced by the forefoil (dorsal/ventral fin), thus supporting the hypotheses of experimental biologists. A recent numerical investigation by Liu et al., used reconstructed geometries and kinematics from video of swimming crevalle jack (*Caranx hippos*) to investigate the effects of the body and fins on caudal fin thrust. They found that including the dorsal/ventral fins could improve the caudal fin thrust by 13.4% when compared to a model with the body and caudal fin only or by 43.2% when compared to an isolated caudal fin [22]. These works suggested that the dorsal/ventral fins have the ability to significantly increase the performance of the caudal fin and the entire fish as a whole.

The importance of body kinematics has also been shown to be important [14,15,16,22,23,33,34,35,36]. Wu analytically found that a flexible two-dimensional panel deforming with a transverse traveling wave is preferable to a rigid panel [33]. This was experimentally supported by Katz and Weihs who found that chordwise flexibility can increase efficiency by up to 20% without significant thrust reduction as compared to a rigid panel [34]. Wolfgang et al., used 3D numerical models to show that fish use flexure of their bodies to control body-generated vortices to increase thrust in collaboration with the caudal fin [35]. Using a nonlinear, inviscid flow solver, Zhu et al., studied the interaction of body-generated and caudal-fin-generated vortices. They found that by coordinating the interaction to be either constructive or destructive, the thrust could be improved or efficiency could be improved, respectively [23]. Schouveiler et al., noted that the high performance of thunniform swimmers cannot be attributed to the caudal fin only but must also include the interaction between the body and the caudal fin [36]. More recent work has investigated body flexibility using the simplified model of a flexible thin foil [14,15,16]. Numerical full fish simulations by Liu et al., found that the inclusion of the body could increase the thrust produced by the caudal fin by 29.8% when compared to an isolated caudal fin [22].

In addition to geometry and simple sinusoidal motion, several works have investigated the effects of nonsinusoidal motion [4,6,37,38,39,40]. Koochesfahani performed some of the earliest work in this area and found that deviating from a sinusoidal motion has significant effect on the wake structure [4]. Read et al., and Kaya et al., found that performance metrics can be improved through the use of nonsinusoidal motion [6,37]. Recent experimental work by Van Buren et al., investigated a wide range of nonsinusoidal motions using Jacobi elliptic functions in order to generate nondimensional scaling laws. They found that performance was primarily dependent on the peak trailing edge velocity [38]. Das et al., furthered this work using numerical simulations of a self-propelled airfoil. They found that maximum speed was attained with a square-like wave while the energetic efficiency was maximized with a triangle-like wave [39]. Qi et al., found that for small amplitude oscillations, altering the waveform from sinusoidal motion toward a square-like waveform improves performance [40].

Together, these works suggested that optimal thrust and/or efficiency for a given fish is due to some combination of body geometry, dorsal/ventral/caudal fin geometry, and body-fin kinematics. The majority of the experimental work has used two-dimensional velocity data or three-dimensional flow visualization. Several numerical studies have been done on three-dimensional models providing a substantial amount of valuable information. This work seeks to understand the underlying phenomenological relationship between the body and the caudal fin using a fundamental approach with a two degree-of-freedom model and three-dimensional velocity data.

## 2. Materials and Methods

### 2.1. Model

The model used in these experiments is based on the shape and dimensional ratios of the yellowfin tuna (*Thunnus albacares*) and has two degrees-of-freedom. The model, with detailed views and descriptions of its actuation, dimensions, and kinematic parameters, is shown in Figure 1 and Figure 2. The ratios were found from publicly available images of yellowfin tuna as well as from Idyll and Sylva [41]. Maximum body width and body height relative to body length as well as body height to caudal fin span were used to establish the large scale proportions. It was then scaled to fit in our tunnel. The model is composed of four main components; head, sting, tail, and caudal fin. The sting is a stainless steel assembly for rigidity and for the minimization of corrosion due to long exposure to water. The head and tail are 3D printed acrylonitrile butadiene styrene (ABS) material. The caudal fin is laser cut from a 1.59 mm thick optically clear acrylic sheet. Other internal parts were made from brass to minimize corrosion.

The sting contains an outer tube (for mounting), middle tube (for tail actuation), and inner shaft (for caudal fin actuation). The outer tube acts as a sting for the model and has an outer diameter of 19.05 mm and wall thickness of 1.65 mm. The head is rigidly mounted to the outer tube of the sting and remains stationary for all kinematics. It is symmetric about the xy and xz planes with a concave posterior surface to mate with the anterior surface of the tail. The tail is composed of three separate pieces for assembly of internal components and assembled using screws. The tail is rigidly mounted to the middle tube of the sting for actuation. The middle tube is inside the outer tube and has an outer diameter of 12.70 mm and wall thickness of 1.65 mm. The caudal fin is mounted to a 6.35 mm diameter brass cylinder at the posterior end of the tail using two screws. This allows for easy removal of the caudal fin without the need to disassemble the rest of the model. The actuation of the caudal fin is done through a pulley system between the posterior brass cylinder and the inner shaft of the sting. The inner shaft has a diameter of 6.35 mm and is attached to a pulley for transferring torque from this shaft to the posterior cylinder for caudal fin actuation. A rubber belt with a nylon core is used to transfer the torque through a series of three pulleys arranged such that two are rigid and the third can be moved to tension the belt during assembly (Figure 1A). The drive belt serpentines around the three internal pulleys, the inner shaft pulley, and the posterior cylinder.

Above the tunnel, the sting is rigidly mounted to a vertical traverse (Dantec Dynamics model 9041T031) for moving the model between datasets. Above the mounting, the middle tube is driven by a DC motor (Faulhaber Series 3863 CR with a 36:1 planetary gearhead) using a plastic roller chain with a gear ratio of 12:23. The inner shaft continues above the middle tube gear and is driven by a second DC motor (Same model with a 20:1 planetary gearhead) through a plastic roller chain with a gear ratio of 12:23. Both DC motors are controlled using a Galil DMC-4123 motion controller. The position of each motor is recorded using an optical encoder with a resolution of 1024 counts per revolution.

### 2.2. Kinematics

Four kinematic motion profiles with the same trailing edge amplitude were created to span a range between pure sinusoidal pitching of the caudal fin and pure sinusoidal pitching of the whole posterior section with the fin acting as an extension of the tail. Each case is governed by three parameters ($\theta_{T,o}$, $\theta_{C,o}$, and ϕ), as shown in Equations (Equation 1) and (Equation 2) in combination with Figure 2B where *c* is the midspan chord length of the caudal fin, $L_{T}$ is the tail length, and a(t) is the time varying trailing edge amplitude. The angular pitching amplitude between the head centerline and the tail centerline, $\theta_{T,o}$, ranges between $0.00^{\circ }$ and $3.63^{\circ }$. The angular pitching amplitude between the tail centerline and the fin centerline, $\theta_{C,o}$, ranges between $0.00^{\circ }$ and $12.90^{\circ }$. The time varying equations of motion for the tail and caudal fin are shown in Equation (Equation 1). The third parameter, ϕ, is the phase offset between $\theta_{T}$ and $\theta_{C}$ where $\theta_{T}$ leads $\theta_{C}$ by ϕ. Several previous studies have found that peak efficiency occurs when $\phi =90^{\circ }$[6,42,43] and Van Buren et al., found that peak thrust occurs when $\phi =30^{\circ }$[43]. A nominal value of $\phi =70^{\circ }$ was used for cases 2, 3, 6, and 7 while cases 1, 4, 5, and 8 do not have a phase offset because either $\theta_{T,o}$ or $\theta_{C,o}$ is zero. The motion was regular enough to allow for kinematic information to be obtained from phase-averaged particle image velocimetry raw image files. The location of the peduncle joint and the trailing edge were visually tracked to obtain the kinematic information in Figure 3 and Table 1.
(1)$$\theta_{T}\left(t\right)=\theta_{T,o}\sin\left(2\pi ft+\phi \right),\;\theta_{C}\left(t\right)=\theta_{C,o}\sin\left(2\pi ft\right)$$
(2)$$a\left(t\right)=L_{T}\sin\left(\theta_{T}\left(t\right)\right)+c\sin\left(\theta_{T}\left(t\right)+\theta_{C}\left(t\right)\right)$$

Ideally, the model’s measured kinematics would be the same as the kinematics prescribed to the motion controller. This was not always true for our experiments due to unintended flexibility in the belt drive system as well as a slight misalignment between the model components and the freestream. The misalignment for each case is shown in Table 1 where $\Delta \theta_{T}$ and $\Delta \theta_{C}$ are the misalignment angles for the tail and caudal fin, respectively. The tail motion agrees well with the prescribed profiles because the actuation is performed through a direct connection between the middle tube of the sting and the tail with minimal slop. The time history of the tail angle, $\theta_{T}$, for case 4, where $\theta_{T,o}=3.44^{\circ }$ and $\theta_{C,o}\approx 0^{\circ }$, is shown in Figure 3A as a representative example of the tail motion for cases 2, 3, 4, 6, 7, and 8. Cases 1 and 5 are not included because they have negligible tail motion. The measured tail angle, $\theta_{T}$, is shown with circles; the black curve is a fitted sinusoidal curve for comparison; and the dot-dashed line is the angular misalignment, $\Delta \theta_{T}$. The fitted sinusoidal curve matches the amplitude and misalignment offset of the measured data with the prescribed phase offset.

The unintended flexibility in the belt drive system caused a kinematic discrepancy in the caudal fin angle, $\theta_{C}$. It became more apparent as $\theta_{C,o}$ increased and was most noticeable in pure caudal fin motion (cases 1 and 5). The time history of the caudal fin angle, $\theta_{C}$, for case 1, where $\theta_{T,o}\approx 0^{\circ }$ and $\theta_{C,o}=12.90^{\circ }$, is shown in Figure 3B as a worst case example for all eight cases. Cases 5 and 8 are included despite having $\theta_{C,o}\approx 0$ because the discrepancy still manifests due to the forces on the fin and belt drive system. The measured caudal fin angle, $\theta_{C}$, is shown with circles; the black curve is a fitted sinusoidal curve for comparison; and the dot-dashed line is the angular misalignment, $\Delta \theta_{C}$. The fitted sinusoidal curve matches the amplitude and misalignment offset of the measured data with the prescribed phase offset. As the fin sweeps from one extreme to the other, the belt stretches on the pressure side (facing the direction of motion) while the suction side becomes loose. During the change in direction at the motion extrema, the loose side begins to tighten, which eventually leads to a stretched belt on that side. The cycle of stretching in the belt caused the nonsinusoidal profile seen in Figure 3B. The kinematic discrepancy was present in all cases and its effects on the vorticity production and organization in the wake are discussed in Section 3.4.

### 2.3. Experimental Methods

The model was tested in a recirculating water tunnel located at the Syracuse Center of Excellence. The test section has a cross-sectional area of 0.60 m by 0.60 m and a length of 2.44 m with a partial free surface. Upstream of the test section, the flow is conditioned by a honeycomb flow straightener, three screens of increasing fineness, and a contraction. These result in an average freestream turbulence intensity of 0.46% and 1.15% for freestream velocities of 81.5 mm/s and 59.5 mm/s, respectively.

Both two-dimensional, two-component (2D2C) and two-dimensional three-component (2D3C) particle image velocimetry were performed simultaneously over two spatial domains (Figure 4B). The 2D3C setup included two PCO.edge 5.5 megapixel cameras (resolution of 2560 × 2160 pixels${}^{2}$) with 35 mm lenses (Canon EF 35/2.0 IS USM) in angular displacement stereo arrangement above the tunnel. The angle between each camera and the normal to the object plane was $30^{\circ }$ at the camera and $22^{\circ }$ at the object plane due to refraction at the air-acrylic and acrylic-water interfaces. The cameras were positioned such that the object plane was 0.30 m above the bottom of the tunnel (vertically centered) and centered 0.30 m from each side wall (horizontally centered). The streamwise position of the domain was such that part of the model caudal fin was visible while maximizing the amount of the wake visible. This arrangement provided a domain of 0.23 m ×0.28 m with a spatial resolution of 1.78 mm in both the *x* and *y* directions. The 2D2C setup included a single camera, of the same make and model, arranged perpendicular to the object plane and above the tunnel. This camera is further upstream than the 2D3C cameras and positioned to view the flow along the laser side of the tail and caudal fin. This arrangement provided a domain of 0.20 m ×0.23 m with a spatial resolution of 1.48 mm in both the *x* and *y* directions.

The flow was seeded using polyamid seeding particles with a mean diameter of 20 μm. To mitigate surface waves an acrylic cover was placed on roughly half of the test section such that all three cameras viewed the object plane through the acrylic rather than the free surface. A Quantel Evergreen 200 mJ Nd-YAG 15 Hz dual cavity laser was used to illuminate the flow. The laser was located beside the tunnel with optics to create a laser sheet centered on the object plane. A schematic of the setup can be seen in Figure 4A. The model kinematics were periodic with a rate of 1.0 Hz. The PIV system, including both 2D2C and 2D3C setups, collected data simultaneously at a rate of 12.5 Hz. This frequency was selected to allow for the data to be phase averaged over 25 phases where every other phase is collected instantaneously and all 25 phases were collected over two physical periods of motion. This method allows for continuous data collection per plane. After data is collected for a given plane, the model is displaced vertically using a traverse mounted above the tunnel. Traversing the model allows for the cameras and laser to be held stationary and calibration is not required for each individual plane. A to-scale depiction of the five data planes is shown in Figure 4C. When collecting data at the 40 mm plane, the negative spanwise tip of the caudal fin is 0.20 m from the bottom of the tunnel. The positive spanwise tip is 0.20 m from the top of tunnel when collecting data at the −40 mm plane.

The raw images were processed using PIVview2C/3C v3.6 by PIVtec. An interrogation window of 32 × 32 pixel${}^{2}$ with an overlap of 16 pixels was used for both the 2D2C and 2D3C setups. The image evaluation was performed using an FFT correlation algorithm with multi-grid interrogation starting with a 96 × 96 pixel${}^{2}$ window. All post-processing was performed in MATLAB. The 2D2C and 2D3C instantanenous datasets were stitched together into a final domain of 0.34 m ×0.26 m with a spatial resolution of 1.78 mm in both the *x* and *y* directions. These were then phase averaged based on the image sequence into 25 phases.

Experiments included two Strouhal number groups (SG), one at St≈0.27 and one at St≈0.37. Four unique body kinematic groups (KG) resulted in the eight cases detailed in Table 1. For each case, the 0 mm plane (midspan), ±20 mm planes, and ±40 mm planes were collected to capture three-dimensional wake dynamics (see Figure 4C). For cases 1 and 4, the ±40 mm planes were not collected due to experimental constraints. Freestream velocities of 59.5 mm/s and 81.5 mm/s were used to obtain the two Strouhal number groups near 0.37 and 0.27, respectively. These freestream velocities yielded Reynolds numbers (Re=UL/ν) based on body length of 17,000 and 23,000, respectively.

## 3. Results

Three-dimensional visualizations of the velocity data at the five planes for cases 5, 6, 7, and 8 are shown in Figure 5 at two representative phases of the model motion profile. Vortex structures are depicted using contours of spanwise vorticity, thresholded for magnitude values less than 5 s${}^{-1}$. We are clearly capturing a number of coherent phenomena in these wakes: boundary layers and discrete vortices along the body and tail portion of the model, coherent vortices being generated by the motion of the anterior, or leading, swept edge of the caudal fin, and strong spanwise vortices and shear layers being generated by and shed from the caudal fin trailing edge. All these structures show some variation in organization or timing among the different kinematic groups, and will be discussed in the following sections.

### 3.1. Effect of the Body-Generated Vortices

The body of the fish model is composed of the stationary head, stationary sting, and pitching tail section. Each component generates a boundary layer and has the potential to generate vortices that can interact with the caudal fin downstream. The Q-criterion, also referred to as *Q*, is an Eulerian scalar used for vortex identification as proposed by Hunt et al., and its definition is shown in Equation (Equation 3). The velocity gradient tensor, ∇u, can be decomposed such that ∇u=S+Ω, where $S=\frac{1}{2}\left[\nabla u+{\left(\nabla u\right)}^{T}\right]$ is the symmetric rate of strain tensor and $\Omega =\frac{1}{2}\left[\nabla u-{\left(\nabla u\right)}^{T}\right]$ is the anti-symmetric rate of rotation tensor. ||Ω|| represents the Euclidean (Frobenius) norm of Ω.
(3)$$Q=\frac{1}{2}{\left(||\Omega |\right|}^{2}-{\left||S|\right|}^{2})$$

Any region with values of *Q* larger than zero are regions where the local rate of rotation, Ω, is dominant over the local rate of strain, *S* [44]. In Figure 6, Figure 7, Figure 8 and Figure 9, the lowest contour is set to $1s^{-2}$, which is approximately 1% of the global maximum value. The value is set above zero to eliminate experimental noise in the visualization. Figure 6A shows the wake beside the body and caudal fin at the midspan plane (0 mm) for case 8 ($\theta_{T,o}=3.63^{\circ }$, $\theta_{C,o}\approx 0^{\circ }$). There are several small structures forming in the boundary layer of the tail, but none of these structures are strong enough to persist downstream and interact with the caudal fin. Figure 6B shows the wake between the body and caudal fin at the +20 mm plane. Small structures in the boundary layer are also visible in this plane, but none of them persist long enough to interact with the caudal fin. Figure 6C shows the wake between the body and caudal fin at the +40 mm plane. We believe that the vortices shown here are generated by the sting, and they appear to dissipate in the flow before being able to interact with the caudal fin as discrete vortices. The sting did generate more turbulence in the flow that caused more uncertainty in the phase-averaged data. For this reason, the two negative planes are used in future sections to describe the flow at the ±20 mm and ±40 mm planes. The flow between the body and caudal fin could not be observed below the midspan plane due to the body shadow (−20 mm and −40 mm planes). Overall, the body-generated vortices are not strong enough to persist in the flow and interact with the caudal fin. This is in agreement with the recent numerical work by Liu et al., who observed that the body-generated vortices of a crevalle jack fish without the dorsal/ventral fins were not strong enough to interact with the caudal fin in a meaningful way [22]. Observations described here are consistent across all eight cases.

### 3.2. Leading Edge Vortex Formation

As the fin moves relative to the surrounding fluid, a shear layer forms along each edge of the fin as fluid flows around the edge. We refer to the anterior swept edge of the caudal fin as the leading edge, and a vortex that forms along this edge as a leading edge vortex (LEV). Borazjani and Dghooghi [24] and Liu et al. [22] have previously identified LEVs forming on the caudal fin of carangiform swimmers. Our results are consistent with those, identifying LEVs using *Q* at the z=−40 mm plane (Figure 7) and z=−20 mm plane (Figure 8). We only present the planes below the midspan (z<0) as they are less affected by the wake of the sting. The LEVs described here are on the +y side of the fin, which is not in the shadow of either the fin or the body/tail. The 0 mm plane does not form an LEV because the body connects to the fin in this region (−6mm<z<6mm), eliminating the leading edge.

During each half-cycle, an LEV forms along the swept leading edge of the fin due to a shear layer whose strength is proportional to the edge’s velocity relative to the surrounding fluid. For all cases, the edge’s velocity is smallest at the peduncle joint and increases along the chord of the rigid fin. This creates an LEV with nonuniform circulation that is weakest at the peduncle joint and strongest near the spanwise tips. Buchholz and Smits [7] linked a nonuniform chordwise edge vortex to a chordwise pressure gradient on the surface of the fin. Simply for illustrative purposes, we use case 7 as an example here to describe the life-cycle of an LEV forming on the caudal fin. Videos of all eight cases for the −20 mm and −40 mm planes (where applicable) are available in Appendix A. The −40 mm and −20 mm planes are shown in Figure 7 and Figure 8, respectively.

The example starts at t/T=0.50 with the caudal fin at the positive amplitude extreme and starting to move downward (between Figure 7A,B). The LEV forms at t/T≈0.52 and can be seen in Figure 7B (t/T=0.64). It continues to grow in size and strength until t/T≈0.80 (Figure 7C). The relative strength of LEVs is determined using *Q* contours to compare the size and magnitudes within the structures. If two structures have similar *Q* values throughout and one is larger than the other, we infer that the larger structure contains more circulation and is considered “stronger”. Similarly, if two structures have similar size and one has higher peak magnitudes of *Q*, then the structure with higher *Q* magnitudes is assumed to contain more circulation and is considered stronger. At t/T=0.80 (Figure 7C), the LEV has already detached from the surface at the ±40 mm planes, but remains attached at the ±20 mm planes (compare Figure 7C and Figure 8C). We consider the LEV to no longer be attached if a gap is present between the lowest contour level and the fin. At the ±40 mm planes, the shear layer feeding the vortex is pinched-off between t/T=0.80 and 0.88, allowing the shed LEV to advect along the fin (Figure 7D,E) and eventually merge with the forming trailing edge vortex (TEV) of the same sign (1.00<t/T<1.20, Figure 7F). After the merging, the trailing edge continues to move upward generating additional vorticity that is not entrained in the primary vortex (Figure 7G,H). At the ±20 mm planes, the LEV remains attached (Figure 8C–E) and does not appear to advect along the fin. Instead, it dissipates, or is swept back around the leading edge as an LEV is formed on the opposite side during the next half-cycle (Figure 8F–H). Some combination of these two behaviors is also possible.

A few trends are observed across the four kinematic groups. The first is a result of the fact that the relative velocity of the leading edge, for a given nondimensional time and distance from the fin’s pitching axis, obviously increases with maximum tail amplitude ($\theta_{T,o}$) as this angle is related to the motion of the peduncle. The size and strength of the LEV increases accordingly as can be seen in Figure 9 (both −40 mm and −20 mm planes). In these plots, case 5 (Figure 9A,E) has the smallest LEV and case 8 (Figure 9D,H) has the largest LEV.

Second, the numerical investigation of crevalle jack fish with video-captured geometry and kinematics by Liu et al., observed that the entire LEV was advected along the fin chord and merged with the TEV. In the current work, and for all cases, this behavior is only observed at the ±40 mm planes while the LEV at the ±20 mm planes appears to dissipate, is swept back across the leading edge, or some combination of these two. For all cases at the ±40 mm planes, the LEV begins to merge with the TEV around the same nondimensional time (t/T=1.00) and continues through t/T=1.20 (mid-merge is shown in Figure 7F).

Third, with increasing maximum tail amplitude the LEV forms along the entire leading edge earlier in the half-cycle. At the ±40 mm planes, the LEV also detaches from the surface earlier in the half-cycle with increasing maximum tail amplitude. For cases 5 and 6, the LEV detaches from the surface between t/T=0.88 and 0.96. For cases 7 and 8, which have larger tail amplitude, the detachment occurs between t/T=0.64 and 0.80. The top row of Figure 9 shows that at t/T=0.88, the LEVs in cases 5 and 6 are still attached (Figure 9A,B) while in cases 7 and 8, they are already detached (Figure 9C,D). We note here that in general, the relative velocity of the leading edge and the strength of the resulting LEV would decrease as phase offset between tail and caudal fin motion (ϕ) increased from $0^{\circ }$ to $180^{\circ }$, if TE amplitude (*A*) were kept constant.

The numerical simulations by Liu et al., linked the presence of attached LEVs to peaks in thrust, showing the significance of LEVs in the generation of thrust [22]. The data shown in the current work suggests that the phenomenon of LEV attachment and detachment is also related to the leading edge velocity, which is governed by the body-fin kinematic parameters ($\theta_{T,o}$, $\theta_{C,o}$, and ϕ). We acknowledge that the kinematic discrepancy (discussed in Section 2.2) may have an effect on the formation and shedding of the LEVs, but the connection has not been determined. We believe that the link is less obvious because only one LEV is formed per half-cycle unlike the TEV where multiple vortices are shed per half-cycle.

### 3.3. Circulation Production in the Wake

The amount of circulation generated per half-cycle at the midspan in these experiments is similar among cases with similar kinematics despite having different freestream velocities. This supports the hypothesis that the magnitude of circulation production is insensitive to freestream velocity as proposed by Buchholz et al. [45]. The first and second Strouhal number groups have a nominal Strouhal number (St) of 0.27 and 0.37, respectively. In our experiments, $U_{\infty }$ was varied to change St. Within each Strouhal number group, St varies slightly due to experimental variation in *A*; but between the two groups, the freestream velocity is the dominant distinguishing parameter.

A time history of circulation produced by the trailing edge at the midspan was calculated by considering a rectangular region downstream of the TE. The region is tall enough to capture the full transverse extent of the wake and extends far enough downstream to contain the total amount of circulation shed in one half-cycle. To do this, the upstream and transverse boundaries of the region are held stationary while the downstream boundary moves with the approximate vortex advection speed. Case 5 is used in Figure 10 as an illustrative example for the method of calculating the circulation history. This figure shows the first phase with vorticity present (Figure 10A, t/T=−0.12), the phase with peak circulation (Figure 10B, t/T=0.32), and a later phase (Figure 10C, t/T=0.64) once positive (red) vorticity is no longer being generated. The phase t/T=−0.12 is before the half-cycle begins and it is shown because it is the beginning of vorticity generation for the given half-cycle. Each half-cycle is defined by the trailing edge reaching an amplitude maximum or minimum. An area integral of the vorticity magnitude above a threshold of $1s^{-1}$ within the region was used to calculate the circulation, and the time history of positive and negative circulation for this case can be seen in Figure 11B as the solid red and blue curves, respectively. The positive and negative circulation time histories for the other seven cases are shown in Figure 11B through Figure 11E. For each case, the circulation rises during the beginning of the half-cycle, reaches a maximum value, and then plateaus. The plateau indicates that the amount of circulation, of that sign, remains relatively constant with a slow decrease due to viscous dissipation and interaction with neighboring structures. The maximum value at the beginning of the plateau region is considered the total circulation produced during that half-cycle.

For each time history (Figure 11B through Figure 11E), the positive and negative circulation produced per half-cycle match reasonably well with some variation during the beginning of the half-cycle. The consistency supports experimental symmetry. The maximum circulations produced per half-cycle within each kinematic group are compared in Figure 11A, and the values align well for kinematic groups 2, 3, and 4. Kinematic group 1 does not match as well between the two Strouhal numbers. One possible explanation for this is that the motion profile in kinematic group 1 is characterized by entirely caudal fin motion and, as discussed in Section 2.2, the kinematic discrepancy is most noticeable in this motion.

The discrepancy-caused variation in the trailing edge amplitude, *A*, among all cases was summarized in Table 1. In each kinematic group, this resulted in a difference between the two St cases, but it was not always true that the larger *A* corresponded to the higher St (recalling that St was set by adjusting the freestream velocity and nominally holding *A* constant). In kinematic groups 2, 3, and 4, the larger trailing edge amplitude between the two cases was 5.7%, 6.4%, and 1.8% higher than the lower one, respectively, while for kinematic group 1 the difference was 13.6%. Of this kinematic group, case 1 (dashed lines) and case 5 (solid lines) are shown in Figure 11B. Despite its lower St, case 1, which had A=25.1 mm, had a larger trailing edge amplitude and produced more circulation than case 5, which had A=22.1 mm. The differences of *A* between the cases in kinematic groups 2, 3, and 4 are smaller than in kinematic group 1, and the time histories of circulation produced are more similar. The fact that there is *not* a large difference between the cases in kinematic groups 2, 3, and 4 indicates that the amount of circulation produced per half-cycle is relatively insensitive to freestream velocity. The large difference in circulation produced between cases 1 and 5 suggests that circulation production is sensitive to *A* and by extension trailing edge velocity. The pitching frequency (*f*) is held constant for all cases so an increase in *A* corresponds to an increase in trailing edge velocity.

### 3.4. Effect of Nonsinusoidal Trailing Edge Kinematics

The vorticity in the wake of the fish model is arranged in a series of vortices that each originates in a shear layer created by the relative motion of the caudal fin trailing edge (TE) and the surrounding fluid. For this reason, the generation and arrangement of vorticity in the wake behind this two degree-of-freedom model is sensitive to the trailing edge kinematics. The experimental data is composed of four kinematic groups, which result in four distinct wake structures. Each will be explained in relation to their TE kinematics. Cases 1 and 4 will be discussed in more depth to exemplify the differences between a square-like (case 1) and a sinusoidal (case 4) waveform.

The TE amplitude is held relatively constant in cases 2 through 4 while case 1 has a TE amplitude that is larger due to experimental error. Despite this, the main difference between the cases is the amount of the TE amplitude that is contributed by either the tail or the caudal fin. To better understand how this affects the motion waveform we consider a simple example (Figure 12) of adding two sinusoids (a model tail motion *T*, dotted curve, and a model caudal fin motion *C*, dashed curve) with a fixed offset between them (ϕ) that results in a third sinusoid (the trailing edge excursion *A*, solid curve). The vertical lines in the figure represent the timing of each peak. ϕ is equal to the distance between the vertical dashed and dotted lines, and is the same in both figures. The phase offset between the model resultant curve (*A*) and the model caudal fin curve (*C*) is the distance between the vertical solid and dashed lines. Figure 12 shows two examples where the resultant curves (*A*) have the same peak amplitude. In Figure 12A, the majority of *A* is from *T*; while in Figure 12B, the majority of *A* is from *C*. From these examples, we can see that the phase offset between *C* and *A* is dependent on the relative peak amplitudes of *T* and *C*. In the experiments, the phase offset between the tail and caudal fin motion (ϕ) is fixed at $70^{\circ }$ and the TE excursion is fixed at approximately 22 mm, but the phase offset between the caudal fin and the TE motion thus varies. The actual waveform of the caudal fin in our experiments is square-like due to the kinematic discrepancy. This additionally causes the waveform of the TE motion to vary depending on how much of the amplitude is contributed by the caudal fin. Both effects are observed to have a significant impact on the vorticity generated at the trailing edge.

The actual experimental waveforms can be seen in Figure 13. In this figure, the TE amplitude is represented by the solid curve and the TE velocity is represented by the dashed curve. These curves are overlaid with colored regions representing time periods of TE acceleration (red) and TE deceleration (blue) as well as yellow circles representing the approximate timing of vortex shedding. The diameter of the circles is equal to the temporal resolution of the dataset. Due to the discrete nature of experimental data, we are unable to pinpoint the exact timing of events, but we can identify a time period during which the event took place. For the purposes of this discussion, the yellow circles will be identified by the center time that occurs between the experimental time steps.

Figure 14 shows the vorticity arrangement for cases 1 through 4 at t/T=1.00. In cases 1 through 3, the primary vortices (P1 and P2) are shed during periods of TE deceleration; while in case 4, the primary vortex is shed during a period when the TE is still accelerating. This suggests that two different shedding mechanisms may be involved. In case 1, the TE is weakly accelerating between t/T=0.50 and 0.60 (Figure 13A). During this time period, the TE is nearly stationary as previously described and results in a weak shear layer that rolls up into discrete vortices due to what we assume are general shear layer instabilities. These early vortices will be referred to as secondary vortices (S1, S2, etc.). The primary vortex grows during a period of TE acceleration (0.60<t/T<0.78) and is shed at t/T≈0.81 during a period of TE deceleration. The full half-cycle can be seen in Figure 15 and the vorticity arrangement at the end of the half-cycle can be see in Figure 14A. In case 2, the first primary vortex (P1) is formed during a period of TE acceleration (0.50<t/T<0.62) and is shed at t/T≈0.66. The second primary vortex (P2) is also formed during a period of TE acceleration (0.68<t/T<0.88) and is shed at t/T≈0.90. Both vortices are shed during periods of TE deceleration (Figure 13B and Figure 14B). In case 3, the first primary vortex (P1) is formed during a period of TE acceleration (0.50<t/T<0.65) and is shed at t/T≈0.66. The second primary vortex (P2) is also formed during a period of TE acceleration (0.75<t/T<0.80) and is shed at t/T≈0.86. Both vortices are shed during periods of TE deceleration (Figure 13C and Figure 14C). In case 4, the primary vortex (P1) is formed during a period of TE acceleration (0.50<t/T<0.73) and is shed at t/T≈0.70 while the remaining shear layer rolls up into trailing secondary vortices (S1 and S2). Unlike the previous three cases where the primary vortices are shed during periods of TE deceleration, the primary vortex in this case is shed during the initial period of TE acceleration. The full half-cycle can be seen in Figure 16. In cases 1 through 3, the deceleration of the TE results in the shedding of a forming vortex while in case 4 we believe the vortex becomes saturated and subsequently sheds during a period of TE acceleration. The mechanism in case 4 may be consistent with DeVoria and Ringuette who suggested that a steadily forming vortex will shed once it has become saturated and any remaining circulation is collected into a series of small vortices in the wake of the primary vortex [46].

A second trend in cases 1 through 4 concerns the distribution of vorticity between the vortices that are shed per half-cycle. In case 1, The TE motion is entirely a result of the caudal fin and the wake consists of a weaker first vortex and a stronger second vortex. As tail motion is added, the first vortex becomes stronger and the second vortex becomes weaker. This is seen in cases 2 and 3 where the case 2 wake has two vortices of similar strength and case 3 has a stronger first vortex and a weaker second vortex. We hypothesize that if more cases are generated between these cases a clear transition will be observed. This transition may not be caused by the addition of tail motion itself, but is more likely the result of two factors: the short periods of deceleration due to the kinematic discrepancy, which weakens as the motion shifts from all caudal fin contribution to all tail contribution; as well as the changing phase offset between the total TE motion and the caudal fin motion.

Case 4 is shown in Figure 16, where St=0.27 and the kinematics are characterized by a large amplitude tail motion ($\pm 3.44^{\circ }$) and negligible caudal fin motion (≈$0^{\circ }$). Phase-averaged TE amplitude and velocity curves versus nondimensional time are shown in Figure 13D. The kinematic discrepancy is barely apparent in this case due to the negligible caudal fin motion, and therefore this case represents nearly ideal sinusoidal trailing edge motion as is used in many previous studies in the literature. At t/T=0.40 (Figure 16A), the TE is decelerating as it approaches the positive amplitude extreme at the beginning of the second half-cycle (0.50<t/T<1.00). The previously generated negative vortices (blue) have been shed while a small amount of positive vorticity (red) has started to form and is already visible at the TE. Between t/T=0.40 and 0.50, the TE decelerates until it reaches the positive amplitude extreme at the beginning of the second half-cycle (Figure 16B). At this time the TE reverses direction and starts to form a strong shear layer. Between t/T=0.50 and 0.75, the fin accelerates to its largest downward velocity while the shear layer continues to feed the attached forming vortex (Figure 16B,C). The velocity is here $90^{\circ }$ out-of-phase with the TE amplitude as expected for an ideal sinusoid and therefore the velocity peak coincides with the TE amplitude zero crossing. At t/T≈0.72 (yellow circle in Figure 13D), the vortex is shed and is visible in Figure 16D. Between t/T=0.75 and 1.00, the TE decelerates as it moves downward toward the negative amplitude extreme at the end of the current half-cycle. During this time interval, the shear layer weakens and part of the remaining vorticity rolls up into a secondary vortex that trails the primary vortex (Figure 16E). At some point before t/T=1.00, the fluid above the fin starts to impinge on its top surface and travel from top to bottom around the TE generating negative vorticity (blue) as seen in (Figure 16F). From there, similar but opposite kinematics and vorticity arrangement are seen in the next half-cycle.

Case 1 is shown in Figure 15, where St=0.31 and the kinematics are characterized by a negligible tail motion (≈0.0${}^{\circ }$) and a large amplitude caudal fin motion ($\pm 12.90^{\circ }$) that exhibits the discrepancy described in Section 2.2. Phase-averaged TE amplitude and velocity curves versus nondimensional time are shown in Figure 13A. At t/T=0.40, the TE is decelerating as it approaches the positive amplitude extreme at the beginning of the second half-cycle (0.50<t/T<1.00). Figure 15A shows that the previously generated negative vortices (blue) have been shed and positive vorticity (red) is visibly being created at the TE as the fin decelerates its upward motion. As in case 4, the fluid below the fin wants to continue moving upward due to its inertia, causing it to impinge on the fin and travel upward around the TE, generating a shear layer. In this case, between t/T=0.40 and 0.64 the TE decelerates but then remains almost stationary at the positive amplitude extreme due to the kinematic discrepancy. Although the fin is almost stationary, the surrounding fluid continues to move upward due to its inertia. The fluid near the TE continues to travel around the TE, forming a relatively weak shear layer that extends linearly into the wake and can be seen in Figure 15B,C. As a result of instabilities in the shear layer, it rolls up into several small vortices (Figure 15C). The largest of these vortices remains coherent and advects downstream following a linear path inclined relative to the *x*-direction (see dashed arrows in Figure 15). Van Buren et al., also observed a small preliminary vortex being formed and shed at the beginning of their square-like profile case [38]. After t/T=0.64, the TE begins to accelerate as it moves downward, generating a stronger shear layer that rolls up to form an attached vortex. At t/T=0.78, the TE reaches its largest negative velocity at the centerline while continuing to generate vorticity that feeds the attached vortex. The TE then begins to decelerate and the primary vortex is shed at t/T≈0.81 (yellow circle in Figure 13A). Figure 15E shows the primary vortex after it has been shed from the TE. The TE continues to decelerate, weakening the shear layer until at sometime between t/T=0.84 and 1.00, the tendency of the fluid on top of the fin to continue downward with its own inertia overcomes the previous flow from bottom to top, and negative vorticity is then generated. The final arrangement of vorticity at the end of the half-cycle is shown in Figure 15F. At this time (t/T=1.00), the negative amplitude extreme has been reached and the next half-cycle begins. From there, similar but opposite kinematics and vorticity arrangement are seen in the next half-cycle.

The two intermediate cases in this Strouhal number grouping (cases 2 and 3) exhibit similar vortex shedding dynamics. A thorough description is not included here for brevity, but associated figures are included in Appendix A, which highlight the vorticity generation and organization for all eight cases. As described, in case 1 a shear layer rolls up into a secondary vortex before the trailing edge begins to accelerate and form the primary vortex. In case 4, the panel begins to accelerate sooner, so the shear layer stays attached and forms the primary vortex first. It saturates and sheds, and the trailing shear layer rolls up into secondary structures. In the intermediate cases, the timing of the TE acceleration does move earlier, but the periods of deceleration due to the kinematic discrepancy cause the first vortex to pinch off before saturation, and leaves time for a stronger secondary vortex. As the discrepancy lessens, this effect lessens.

It is deduced that the kinematic discrepancy is not only affecting the magnitude and distribution of vorticity shed from the trailing edge, but likely also the forces generated as well. The time-averaged normalized streamwise velocity for cases 1 through 4 are shown in Figure 17 while cases 5 through 8 are shown in Appendix A. This quantity is used to visualize regions where momentum has been added to the wake to produce thrust ($U/U_{\infty }-1>0$, red) and regions of momentum deficit (blue). It is clear that the segmentation of the primary vortex in case 2 (Figure 17B) cases a bifurcation in the higher velocity jet generated in the wake, and from both the smaller area of momentum surplus, and its relatively lower magnitude, we may assume that a lower force was generated as well. There is also some jet bifurfaction seen in case 1. Cases 3 and 4 exhibit a coherent jet core in the midspan plane, and the magnitude and area is largest in case 4. While we are not able to infer anything specific about the force and efficiency of the fish model, the connections are apparent among body kinematics, trailing edge velocity, vorticity production and organization, and momentum organization.

## 4. Discussion

A two degree-of-freedom experimental model of simplified generic fish body swimming motion was designed and built for water tunnel experiments. A combination of two-component and three-component phase-averaged PIV was acquired around the posterior of the model and in its wake. The model was actuated to achieve similar trailing edge motions with different combinations of tail and caudal fin contributions, to investigate the effects of the body kinematics on the wake structure, and potentially performance, of the modeled swimmer.

A particular discrepancy in the actual motion of the trailing edge, compared with the expected ideal sinusoidal motion, was noted and described. While not intended, the results were informative. Pure sinusoidal motion is commonly used in the bio-inspired fluid dynamics field as models of fin motion, but from these results we learned that relatively minor deviations from a sinusoidal motion profile can result in large-scale changes in the organization and shedding of wake vortex structures. Considering that real fish and aquatic mammals likely do not articulate their fins in perfectly periodic sinusoids, we pursued an inspection of what features of the kinematic discrepancy actually led to the changes in wake structure. This understanding may lend insight into how applicable lab results are, and give some guidance to experimental strategy in the future. It is important to note that our observations are consistent with others in the literature who have studied changing motion profiles in particular, and found that performance metrics can be optimized by manipulating the waveform of an airfoil undergoing pitch and/or heave [6,37]. More recent studies by Das et al. [39] and Van Buren et al. [38] have shown that thrust is maximized with a square-like waveform while efficiency is maximized with a sinusoidal [38] or triangular [39] waveform. In the work presented here, we present a more detailed look at the flow field evolution due to particular features of the body motion, and how that may be contributing to the changes in performance. This understanding could lead to generalized insights about how to design more powerful or efficient swimming motions.

The following observations were made and described in detail: vorticity creation around the body itself, the creation and dynamics of vortices generated around the leading edge of the caudal fin, the amount of circulation generated, and the generation and organization of vortices at the caudal fin trailing edge.

Previously, Liu et al., found that the inclusion of the body increased the thrust of the caudal fin when compared to an isolated caudal fin undergoing the same motion. In their work and the work we presented here, the body alone does not generate coherent vortices that advect downstream and interact with the caudal fin. The presence of sharp trailing edges on either the body or dorsal/ventral fins would likely be required. The benefit to the performance of the swimmer is then assumed to be related to the overall pressure field around the body surface, or another yet unexplained phenomenon.

At the leading edge, attached leading edge vortices (LEVs) are known to create low pressure zones on wing surfaces and would increase thrust if the surface is angled appropriately with the low pressure side facing upstream, which occurs during the early stages of the half-cycle. Liu et al., linked the presence of stable attached LEVs to peaks in thrust [22], and Anderson et al., and Read et al., both found that thrust is optimized when the LEV and trailing edge vortices coalesce in a beneficial way. In our results, it was observed that the leading edge vortex (LEV) forms earlier in the half-cycle when the maximum tail (as opposed to caudal fin) amplitude is increased (from case 1 to 4 or case 5 to 8). The LEVs also detach from the surface earlier in the cycle (with increased maximum tail amplitude) before the orientation of the fin is reversed and the low pressure zone causes drag.

While direct time-resolved thrust measurements were outside the scope of this work, it is clearly seen that the body kinematics do have specific large-scale effects on the wake, from which we can infer changes in performance as well. In addition to the timing of LEV generation and shedding, the changing kinematics were shown in this paper to have a strong effect on the organization of vorticity shed from, and amount of circulation generated at the trailing edge. In Section 3.4, it was shown that across the kinematic groups, different numbers of vortices were shed at different times in the model motion, and traveled along different trajectories in the wake. The circulation generated at the midspan was significantly higher for one of the cases, despite having similar Strouhal numbers, freestream velocities, and trailing edge amplitudes. Lastly, the organization of momentum surplus, used as a way to determine where and how the thrust generated by the model motion added momentum into the wake, could have entirely different structure and magnitude across the four kinematic groups (and was consistent with the vortex dynamics).

Lastly, it was noted across the results here, and elsewhere in the literature [12,47,48], that some of the significant trends in unsteady flapping-like fluid dynamics applications are relatively insensitive to the freestream velocity, but particularly sensitive to the trailing edge velocity. This would imply that the Strouhal number, which is commonly used as the most relevant non-dimensional parameter, may indeed not be the most effective when also considering changing motion kinematics.

## Figures and Tables

**Figure 1 biomimetics-04-00067-f001:**
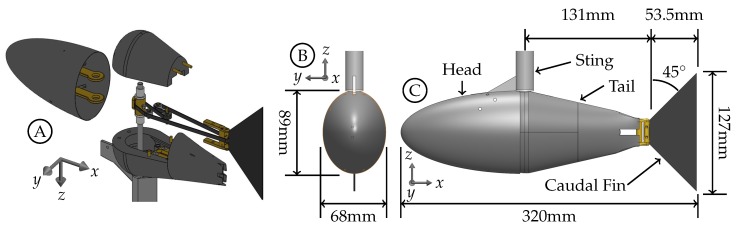
(**A**) Exploded view of the model showing the drive system. (**B**) Front view of model showing dimensions. (**C**) Side view of model showing dimensions and main components.

**Figure 2 biomimetics-04-00067-f002:**
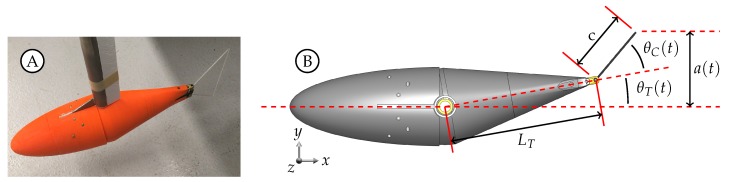
(**A**) Image of the model. (**B**) Schematic depicting the kinematic parameters.

**Figure 3 biomimetics-04-00067-f003:**
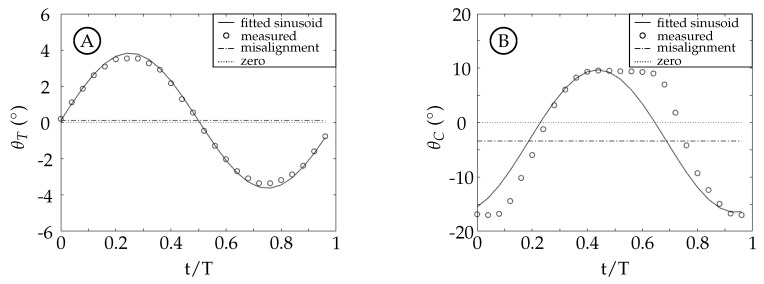
Tail and caudal fin angle for two representative cases. Circles represent the measured angles; the black curve is a fitted sinusoidal curve for comparison; the dotted line is at zero; and the dot-dashed line is the angular misalignment, $\Delta \theta_{T}$ in (**A**) and $\Delta \theta_{C}$ in (**B**). (**A**) Tail angle, $\theta_{T}$, for case 4. (**B**) Caudal fin angle, $\theta_{C}$, for case 1.

**Figure 4 biomimetics-04-00067-f004:**
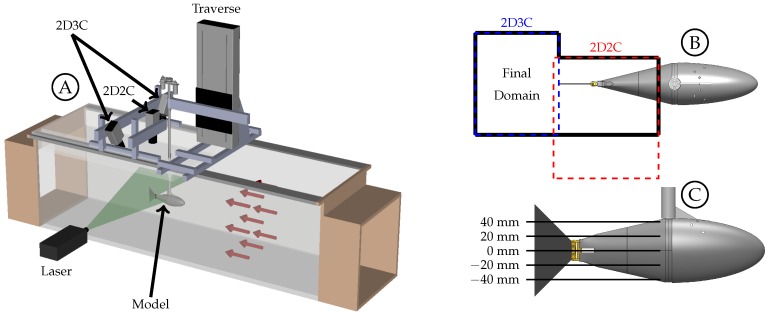
(**A**) Schematic of the water tunnel and experimental setup. (**B**) Top view of the domain relative to the model. (**C**) Five planes where data was collected.

**Figure 5 biomimetics-04-00067-f005:**
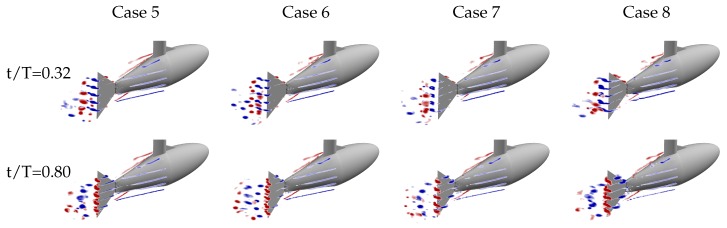
Five planes of spanwise vorticity data for the second kinematic group (cases 5, 6, 7, and 8). Positive spanwise vorticity is shown in red and negative in blue. The top row is at t/T=0.32 where the TE is moving out of the page and highlights the trailing edge vortex. The bottom row is at t/T=0.80 where the TE is moving into the page and highlights the leading edge vortex.

**Figure 6 biomimetics-04-00067-f006:**
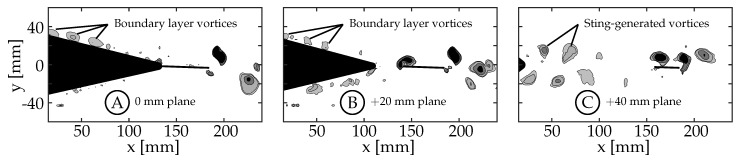
Three planes for case 8, where $\theta_{T,o}=3.63$ and $\theta_{C,o}\approx 0$, are shown here. Body (and sting) generated vortices between the body and the caudal fin are visualized using two-dimensional Q-criterion (*Q*) contours ($Q=\left[1,5,20,50\right]s^{-2}$). (**A**) midspan plane (0 mm). (**B**) +20 mm plane. (**C**) +40 mm plane.

**Figure 7 biomimetics-04-00067-f007:**
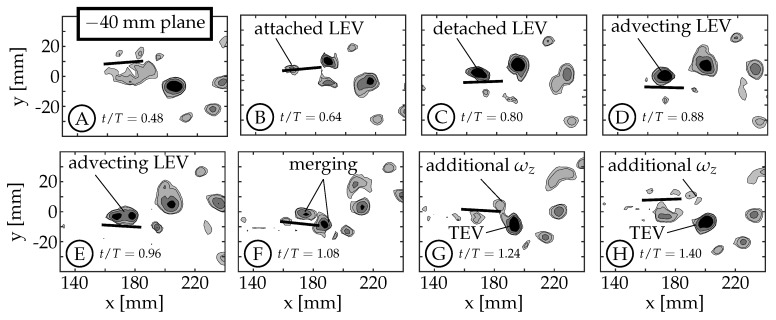
Two-dimensional Q-criterion (*Q*) contours (Q=[1,5,20,50] s${}^{-2}$) are used to identify leading edge vortices during the second half-cycle. These plots show the typical life-cycle of a caudal fin LEV for case 7 at the −40 mm plane between t/T=0.48 and 1.40.

**Figure 8 biomimetics-04-00067-f008:**
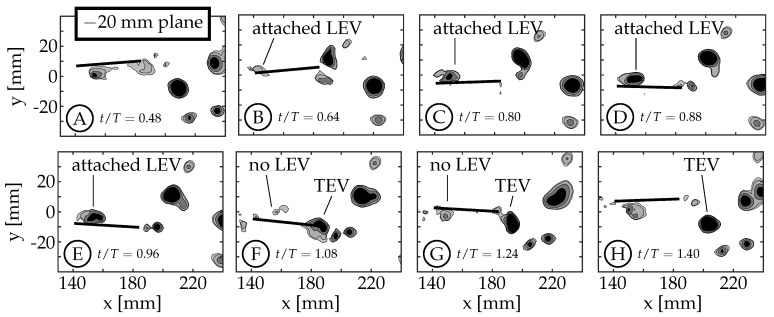
Two-dimensional Q-criterion (*Q*) contours (Q=[1,5,20,50] s${}^{-2}$) are used to identify leading edge vortices during the second half-cycle. These plots show the typical life-cycle of a caudal fin LEV for case 7 at the −20 mm plane between t/T=0.48 and 1.40.

**Figure 9 biomimetics-04-00067-f009:**
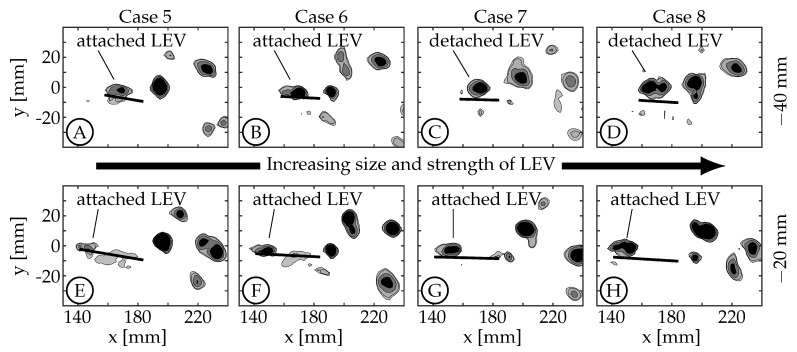
Two-dimensional Q-criterion (*Q*) contours (Q=[1,5,20,50] s${}^{-2}$) are used to identify leading edge vortices during the second half-cycle. Cases 5 through 8 are shown when t/T=0.88. The −40 mm plane is shown in (**A**–**D**) and the −20 mm plane is shown in (**E**–**H**). The size and strength of the LEV increases with increasing maximum tail amplitude.

**Figure 10 biomimetics-04-00067-f010:**
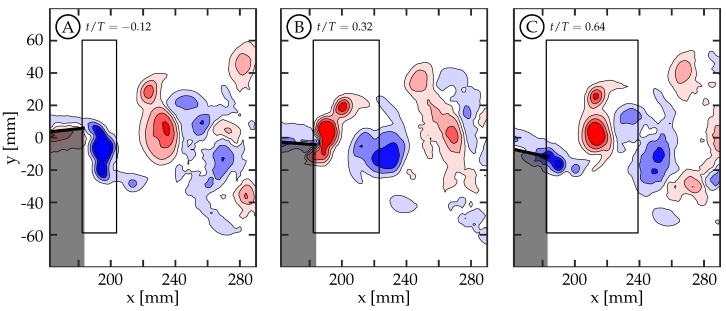
Rectangular region used for calculating positive circulation for case 5. Vorticity contours of $\omega_{z}=\pm \left[1,4,9,16\right]$ s${}^{-1}$ where positive values are shown in red and negative values in blue. The time history for this case can be seen in Figure 11B as the solid red curve. (**A**) t/T=−0.12. (**B**) t/T=0.32 (peak positive circulation). (**C**) t/T=0.64.

**Figure 11 biomimetics-04-00067-f011:**
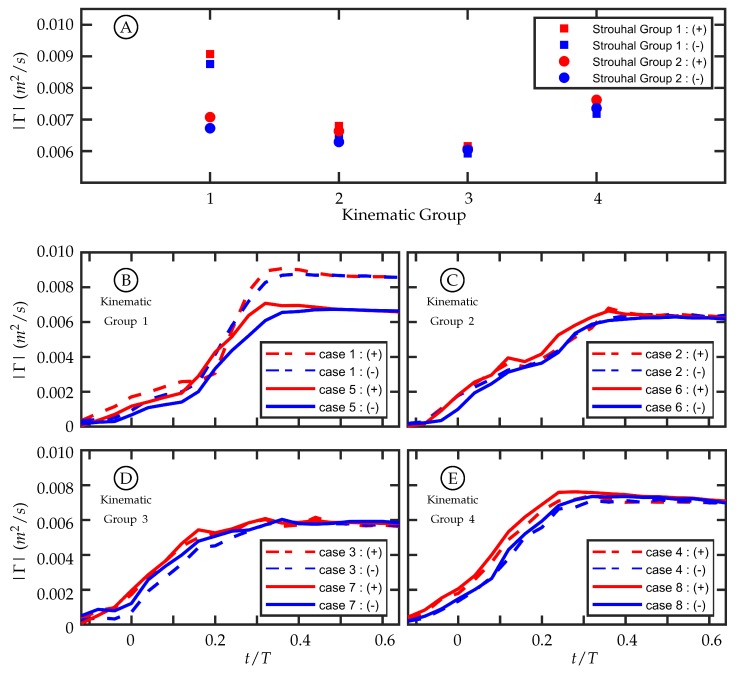
Circulation magnitude of both positive (red) and negative (blue) circulation versus nondimensional time over a pitching half-cycle. (**A**) Total same-sign circulation shed per half-cycle by kinematic group. (**B**) Kinematic Group 1 ($\theta_{T,o}\approx 0^{\circ }$ and $\theta_{C,o}\approx 11^{\circ }$). (**C**) Kinematic Group 2 ($\theta_{T,o}\approx 2^{\circ }$ and $\theta_{C,o}\approx 9^{\circ }$). (**D**) Kinematic Group 3 ($\theta_{T,o}\approx 3^{\circ }$ and $\theta_{C,o}\approx 5^{\circ }$). (**E**) Kinematic Group 4 ($\theta_{T,o}\approx 3.6^{\circ }$ and $\theta_{C,o}\approx 0^{\circ }$).

**Figure 12 biomimetics-04-00067-f012:**
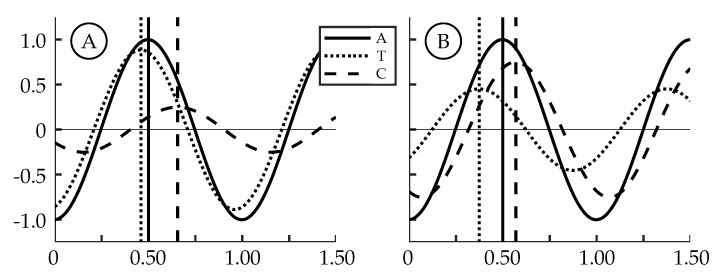
Sample sinusoid summation showing the phase offset between the trailing edge excursion (*A*) and the caudal fin (*C*) motion: (**A**) Motion mainly due to the tail (*T*). (**B**) Motion mainly due to the caudal fin (*C*).

**Figure 13 biomimetics-04-00067-f013:**
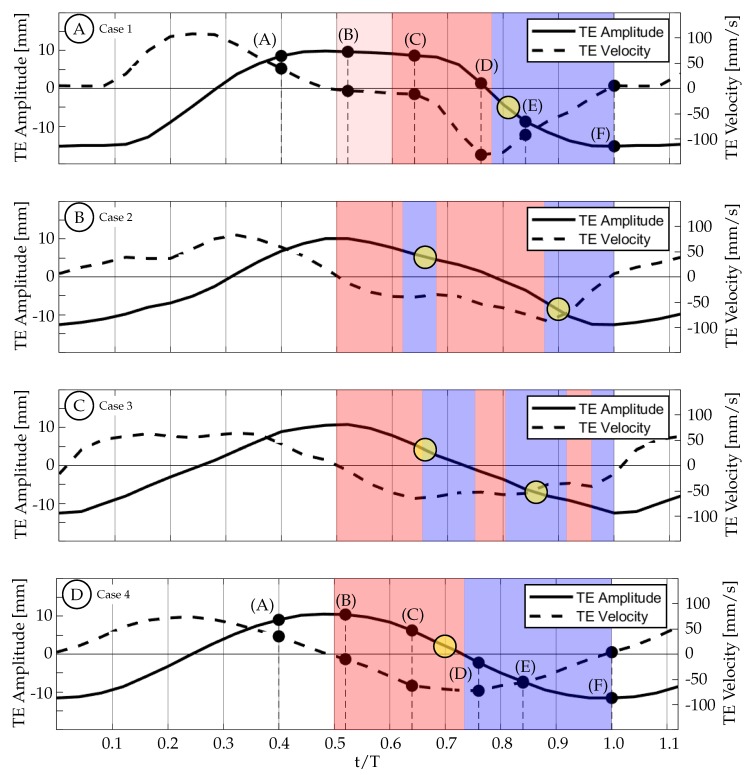
Motion Profiles for cases 1 through 4 where the solid curve represents the trailing edge amplitude and the dashed line represents the trailing edge velocity. The background colors represent time periods of acceleration (red) and deceleration (blue) the yellow circles represent the approximate timing of primary vortex shedding. (**A**) Case 1. (**B**) Case 2. (**C**) Case 3. (**D**) Case 4.

**Figure 14 biomimetics-04-00067-f014:**
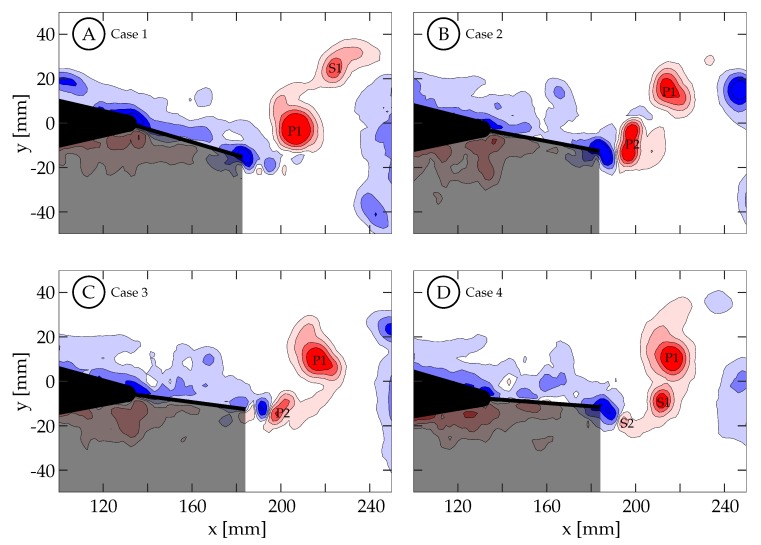
Spanwise vorticity ($\omega_{z}=\pm \left[1,4,9,16\right]$ s${}^{-1}$) contours are shown here for cases 1 through 4 when t/T=1.00. Positive spanwise vorticity is shown in red and negative in blue: (**A**) Case 1. (**B**) Case 2. (**C**) Case 3. (**D**) Case 4.

**Figure 15 biomimetics-04-00067-f015:**
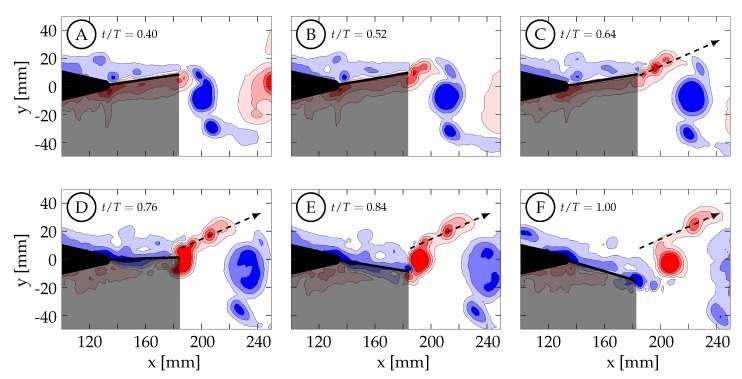
(Case 1: SG1, KG1) Spanwise vorticity ($\omega_{z}=\pm \left[1,4,9,16\right]$ s${}^{-1}$) contours are shown here for case 1 which has $\theta_{T,o}\approx 0.00^{\circ }$, $\theta_{C,o}=12.90^{\circ }$, and St=0.31. Positive spanwise vorticity is shown in red and negative in blue. The dashed arrow highlights the linear trajectory of the secondary vortex. The trailing edge motion profile can be found in Figure 13A: (**A**) t/T=0.40. (**B**) t/T=0.52. (**C**) t/T=0.64. (**D**) t/T=0.76. (**E**) t/T=0.84. (**F**) t/T=1.00.

**Figure 16 biomimetics-04-00067-f016:**
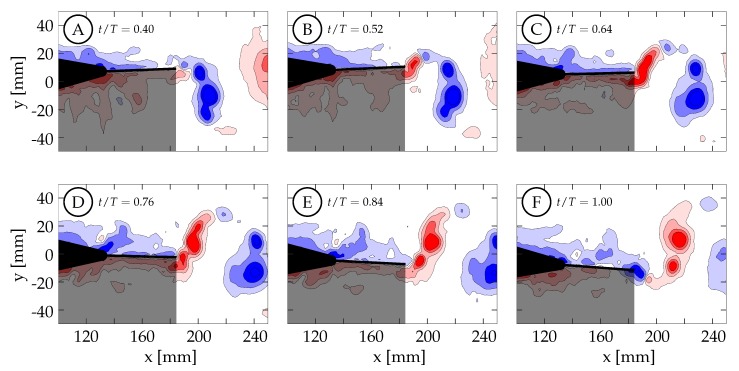
(Case 4: SG1, KG4) Spanwise vorticity ($\omega_{z}=\pm \left[1,4,9,16\right]$ s${}^{-1}$) contours are shown here for case 4 which has $\theta_{T,o}\approx 3.44^{\circ }$, $\theta_{C,o}\approx 0^{\circ }$, and St=0.27. Positive spanwise vorticity is shown in red and negative in blue. The trailing edge motion profile can be found in Figure 13A: (**A**) t/T=0.40. (**B**) t/T=0.52. (**C**) t/T=0.64. (**D**) t/T=0.76. (**E**) t/T=0.84. (**F**) t/T=1.00.

**Figure 17 biomimetics-04-00067-f017:**
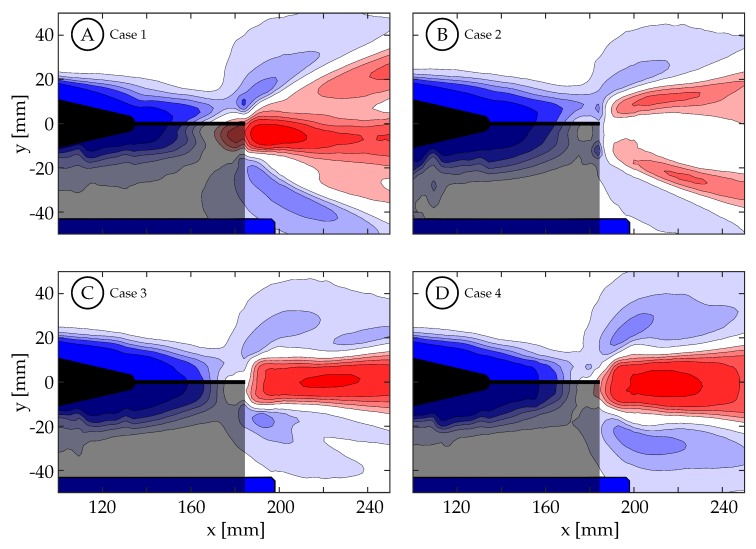
Time averaged *x*-direction velocity ($U/U_{\infty }-1=\pm \left[0.05,0.10,0.15,0.20,0.30\right]$) contours are shown here for cases 1 through 4 where velocity surplus is red and velocity deficit is blue: (**A**) Case 1. (**B**) Case 2. (**C**) Case 3. (**D**) Case 4.

**Table 1 biomimetics-04-00067-t001:** Kinematic parameters for each of the cases investigated. **SG** is the Strouhal number group and **KG** is the kinematic group. $\pm \theta_{T,o}$ is the measured tail angle amplitude with a misalignment of $\Delta \theta_{T}$. $\pm \theta_{C,o}$ is the measured caudal fin angle amplitude with a misalignment of $\Delta \theta_{C}$. ϕ is the phase offset with positive ϕ representing the tail leading the caudal fin. *A* is the maximum excursion of the trailing edge. St is the Strouhal number where $St=fA/U_{\infty }$.

Case	SG	KG	$\theta_{T,o}{(}^{\circ })$	$\Delta \theta_{T}{(}^{\circ })$	$\theta_{C,o}{(}^{\circ })$	$\Delta \theta_{C}{(}^{\circ })$	$\phi {(}^{\circ })$	A(mm)	$u_{\infty }\left(\frac{\mathrm{mm}}{s}\right)$	$f(s^{-1})$	St
1	1	1	0.24	0.09	12.90	−3.4	-	25.1	81.5	1.0	0.308
2	1	2	2.01	0.20	9.03	−2.1	70	22.7	81.5	1.0	0.279
3	1	3	3.08	0.18	5.48	−1.3	70	23.3	81.5	1.0	0.286
4	1	4	3.44	0.09	0.18	−0.9	-	22.1	81.5	1.0	0.271
5	2	1	0.22	−0.07	11.18	−1.5	-	22.1	59.5	1.0	0.371
6	2	2	2.06	−0.04	9.12	−0.7	70	24.0	59.5	1.0	0.403
7	2	3	3.04	−0.09	4.91	0.0	70	21.9	59.5	1.0	0.368
8	2	4	3.63	−0.12	0.68	−1.0	-	22.5	59.5	1.0	0.378

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
