# Peer review of "Experimental Study of Body-Fin Interaction and Vortex Dynamics Generated by a Two Degree-Of-Freedom Fish Model"

_biomimetics, 2019, doi:10.3390/biomimetics4040067_

Round 1
Reviewer 1 Report
1. The references in the Introduction are too old, except for those which contained the basic researches, the others should be replaced by some newer articles, and otherwise the progresses in this field would be lack.
2. There were some clerical errors in the manuscript, such as in the line 114: Both degrees-of-freedom are actuated through the control shafts in th sting; and in the line 440: Between t/T = 84 and 1.00. The authors should check the whole manuscript and revise the errors.
3. From line 114 to line 119, the structure parameters and the functions of the sting were introduced. However, this part could be merged with the paragraph beginning from line 105, for their similar content.
4. The symbols in Figure 3 were explained in the 2nd paragraph and the 3rd paragraph of section 2.2. However, I think it is better to introduce the symbols in the pictures with legends.
5. In Figure 4 A, the devices of the experimental setup should be marked in the picture, which will help readers to understand more clearly.
6. For Figure 5, 9, 11, 12, 13 and 14, the indications of time should be added beside the pictures, so that readers could read easier.
7. Figure 6 C showed the wake between the body and caudal fin at the +40 mm plane, and the vortices were generated by the sting. If the sting is designed as streamline, would the vortices disappear?
8. Figure 7 and 8 showed the generation of LEV in a half-cycle. There were too many pictures in Figure 7 and 8. It is better to choose a group of typical pictures to analyze.
9. Case 7 was an example used to introduce the LEV generation process, while other cases were different with Case 7 at some moments, please summarize the regulations and account for the reasons.
10. Section 3.4 described the formation and evolution of vortices of Case 1, 2, 3 and 4. This section was a little expatiatory, please simplify this part.
Reviewer 2 Report
In this paper the authors present an experimental study of the vortices that are created by body-fin interactions in a model of the tuna’s peduncle and caudal fin. The model has two actuated degrees of freedom, with control over the amplitude, frequency, and velocity of the peduncle and caudal fin. The caudal fin is triangular and presumably rigid (2mm acrylic). This does not model the tuna well but is sufficient for the goals of this study. The model is tested in a flow tank and extensive PIV data are captured. The PIV work is very well done and described thoroughly in the results. The paper is written clearly and its relevance with respect to the literature is described nicely.
I have three concerns with the paper that I recommend be addressed before this paper is published. 1) In section 2.2, the authors reveal that the caudal fin did not execute the prescribed kinematics well. Although I do not think this had a strong impact on the PIV results, I cannot be certain. Why was the model not tested beforehand and fixed prior to conducting these experiments? Were tests conducted subsequently to show that the deviations in kinematics did not affect the results? This should be done. 2) The defined test conditions are not defined with respect to the actual swimming performance of the model. I do understand that the authors wanted to test through a range of ST#, but it would be nice to know how the conditions related to the resultant swimming speed or force production of the system. I believe it is important to know if the vortex structures described in the results relate to the system producing net positive or negative thrust. Do the motions slow the body down, accelerate it, or maintain the steady swimming speed. Please address this so context can be given to the flow structures, which likely would change considerably if the model moved in the direction of its net force production. 3) Finally, please provide a more comprehensive discussion of the results. The discussion section is more focused on what was done than on the impact and meaning of the results. What are the important takeaways of the study? How do these relate to the other studies the author's have identified (line 140). The results are dense, and admittedly I would not want to memorize them to understand their relevance to another type of system. In providing more conclusions, this paper could have relevance beyond those enthused by fluid structures.
Round 2
Reviewer 1 Report
The revised paper has addressed all my previous comments, I suggest to Accept the paper as it is now.